# Thermodynamic Determinants in Antibody-Free Nucleic Acid Lateral Flow Assays (AF-NALFA): Lessons from Molecular Detection of *Listeria monocytogenes*, *Mycobacterium leprae* and *Leishmania amazonensis*

**DOI:** 10.3390/biom15101404

**Published:** 2025-10-02

**Authors:** Leonardo Lopes-Luz, Paula Correa Neddermeyer, Gabryele Cardoso Sampaio, Luana Michele Alves, Matheus Bernardes Torres Fogaça, Djairo Pastor Saavedra, Mariane Martins de Araújo Stefani, Samira Bührer-Sékula

**Affiliations:** 1Instituto de Patologia Tropical e Saúde Pública, Universidade Federal de Goiás, Goiania 74605-050, Brazil; paulacorrea@discente.ufg.br (P.C.N.); gabryele.cardoso@discente.ufg.br (G.C.S.); luanaalves@discente.ufg.br (L.M.A.); matheusfogaca@ufg.br (M.B.T.F.); djairopastor@discente.ufg.br (D.P.S.); mmastefani@gmail.com (M.M.d.A.S.); 2Innovation Hub in Point of Care Technologies, Universidade Federal de Goiás-Merck S/A Alliance, Goiania 74605-050, Brazil

**Keywords:** lateral flow tests, DNA hybridization, Gibbs free energy, pathogen detection, mismatches, single-nucleotide polymorphism, point-of-care DNA Detection

## Abstract

Antibody-free nucleic acid lateral flow assays (AF-NALFA) are an established approach for rapid detection of amplified pathogens DNA but can yield inconsistent signals across targets. Since AF-NALFA depends on dual hybridization of probes to single-stranded amplicons (ssDNA), site-specific thermodynamic (Gibbs free energy-ΔG) at probe-binding regions may be crucial for performance. This study investigated how *site-specific*-ΔG and sequence complementarity at probe-binding regions determine Test-line signal generation, comparing native and synthetic amplicons and assessing the effects of local secondary structures and mismatches. Asymmetric PCR-generated ssDNA amplicons of *Listeria monocytogenes*, *Mycobacterium leprae*, and *Leishmania amazonensis* were analyzed in silico and tested in AF-NALFA prototypes with gold-labeled thiol probes and biotinylated capture probes. T-line signals were photographed, quantified (ImageJ version 1.4k), and statistically correlated with *site-specific*-ΔG. While native ssDNA from *M. leprae* and *L. amazonensis* failed to produce AF-NALFA T-line signals, *L. monocytogenes* yielded strong detection. *Site-specific*-ΔG below −10 kcal/mol correlated with reduced hybridization. Synthetic oligos preserved signals despite structural constraints, whereas ~3–4 mismatches, especially at capture probe regions, markedly impaired T-line intensity. The performance of AF-NALFA depends on the synergism between thermodynamic accessibility, *site-specific*-ΔG-induced site constraints, and sequence complementarity. Because genomic context affects hybridization, target-specific thermodynamic in silico evaluation is necessary for reliable pathogen DNA detection.

## 1. Introduction

Nucleic acid amplification tests (NAATs) are widely used for pathogen detection due to high sensitivity and specificity [1]. However, result confirmation still requires complementary equipment-based methods, such as fluorescence [2], agarose gel electrophoresis [3] or real-time amplification systems and associated software [4]. These methods demand technical expertise, are time-consuming and costly, limiting use in low-resource settings [5]. On the other hand, nucleic acid lateral flow (NALF) allow naked-eye detection of amplicons, emerging as promising point-of-care testing (POCT) tools [6,7,8]. NALF based on lateral flow immunochromatography, called nucleic acid lateral flow immunoassay (NALFIA), relies on antibody–antigen interactions with end-labeled amplicons, commonly conjugated to colloidal gold nanoparticles (AuNPs), similar to traditional POCTs [5]. However, NALFIA faces limitations, including matrix interference in complex samples (notably for foodborne pathogens) [9], amplicon contamination, non-specific binding from primer heterodimers, and signal reduction from unreacted labeled primers or hook effect [10]. Limited availability of target-specific antibodies further restricts multiplex use and raises costs.

As an alternative, the antibody-free nucleic acid lateral flow assay (AF-NALFA) is based on the double hybridization of oligonucleotide probes with the target amplicon [6,11]. This method can be used for the specific detection of genes associated with antimicrobial resistance [12], genotyping and the identification of strains involved in outbreaks [13] as the probes are designed exclusively for a specific region of the single-stranded amplicon (ssDNA) generated by asymmetric amplification. The general strategies for the principle of nucleic acid lateral flow assays rely on the hybridization between a labeled probe and its complementary target sequence, forming a stable duplex that can be immobilized and visualized on the membrane. The efficiency of this hybridization step is therefore the key determinant of detection performance across different NALF formats [6,14]. Over the years, progress in these technologies has evolved to incorporate methods such as Recombinase Polymerase Amplification (RPA) [15], rolling circle amplification (RCA) [16], Loop-Mediated Isothermal Amplification (LAMP) [17], and more recently CRISPR-based detection strategies [18], broadening their potential for rapid and sensitive molecular diagnostics. Despite these advances, the design and optimization of basic components, particularly probe design, remain critical challenges that directly affect detection efficiency and reliability in NALF systems.

Our research group has long-standing expertise in the development of rapid molecular tests, including an AF-NALFA for *Listeria monocytogenes* DNA detection with high specificity and sensitivity [14]. Encouraged by these successful results, we sought to adapt the same platform for *Mycobacterium leprae* and *Leishmania amazonensis*, following the exact same protocols. However, weak visual results for these two pathogens were obtained, despite successful amplification. This discrepancy prompted a fundamental question: why did the *Listeria* assay work so well while the others failed, despite all assays being designed and performed under the same parameters? Our hypothesis was that the target pathogens' sequences generated during asymmetric PCR have not been properly detected in the rapid test.

The dual-hybridization principle of AF-NALFA requires amplicons to bind both capture and detection probes [14]. If one site is structurally inaccessible, hybridization fails, like hidden epitopes in immunoassays. Thus, it is necessary to evaluate the thermodynamic determinants involved in the hybridization between probe and target [19,20]. Thermodynamic determinants of probe–target hybridization include parameters such as Gibbs free energy (ΔG), melting temperature (Tm), base composition, secondary structure stability (hairpins, self-dimers) and the presence of mismatches. These factors collectively govern the accessibility and stability of probe-binding regions. Regarding Tm, which is a common determinant in many hybridization assays, in AF-NALFA the hybridization temperature is controlled by the composition of the running buffer. As previously reported by our group [14], this buffer enables probe-target interactions to occur efficiently at room temperature, neutralizing the influence of Tm on hybridization efficiency.

The advantage of considering the thermodynamic determinants relevant to AF-NALFA, especially at the site-specific level, is that it allows a more accurate prediction of the actual hybridization efficiency, surpassing the conventional analysis based only on the global ΔG. Therefore, we investigated the site-specific Gibbs free energy (*site-specific*-ΔG) of probe-binding regions, proposing that unfavorable local thermodynamics may critically affect AF-NALFA performance. This investigation was based on two hypotheses: (1) inaccessible probe-binding sites due to stable secondary structures, self-complementary interactions or mismatches hinder AF-NALFA T-line signal generation and (2) thermodynamic analysis restricted to the probe-binding regions (*site-specific*-ΔG) can provide a more accurate prediction of assay performance than the overall ΔG calculations for the entire amplicon.

Overall ΔG analysis is a standard method to assess oligonucleotide thermodynamics and is widely used in hybridization assays, such as fluorescence in situ hybridization (FISH) and Southern blot, to ensure efficient probe-target pairing [21,22]. Beyond overall ΔG, hybridization efficiency in ssDNA capture assays depends on secondary structure stability and target accessibility, since highly negative ΔG can favor hairpins and dimers, reducing probe binding [20,23]. In AF-NALFA, however, the overall ΔG does not reflect accessibility at the specific capture and detection sites. Because AF-NALFA relies on dual site-specific hybridization, an assessment of the ΔG of these probe-binding regions is essential to accurately predict assay performance. To our knowledge, no previous studies have specifically addressed this aspect for lateral flow assays in the literature.

This study investigated the thermodynamic determinants of AF-NALFA performance by focusing on the *site-specific*-ΔG and sequence complementarity of the probe-binding regions in asymmetric PCR-generated amplicons. Specifically, we aimed to (i) compare *site-specific*-ΔG profiles among successful (*L. monocytogenes*) and low-performance (*M. leprae* and *L. amazonensis*) assays, (ii) assess the predictive value of *site-specific*-ΔG compared to overall ΔG in determining hybridization efficiency, and (iii) evaluate the influence of secondary structure formation and mismatches within the capture and detection probe-binding regions. To address these aims, we designed truncated and synthetic oligonucleotides based on the respective target genes to better control thermodynamic parameters.

## 2. Materials and Methods

### 2.1. Thermodynamic Algorithms

To evaluate the thermodynamic behavior of the single-stranded DNA (ssDNA) amplicons generated by asymmetric PCR (aPCR) and synthesized, the Geneious^®^ 8.1.9 software (Build 2016-09-27, Java Version 1.7.0_51-b13) and the OligoAnalyzer™ tool from Integrated DNA Technologies (IDT, available at: https://www.idtdna.com/pages/tools/oligoanalyzer, accessed on 1 April 2025) were used. The structural prediction of hairpins formed by the ssDNA was performed using the DNA Fold tool in Geneious^®^, which utilizes the RNAfold folding prediction algorithm from the Vienna package. The energy model was based on Mathews (2004) [24], using a temperature of 25 °C, coaxial stacking was applied as the setting for dangling ends, and the optimal ΔG (both optimal and ensemble) was expressed in kcal/mol.

In Geneious^®^ DNA Fold, the color scheme used to visualize predicted DNA secondary structures is based on the probability of base pairing at each nucleotide position, calculated using the thermodynamic model of Mathews et al. (2004) [24] or an alternative model selected by the user. Warmer colors indicate higher pairing probabilities, while cooler colors represent lower probabilities. The color classification (%) is defined as follows: blue (0.0), cyan (0.1), dark green (0.2–0.3), light green (0.4–0.5), yellow (0.6), light orange (0.7), dark orange (0.8–0.9), and red (1.0). These values reflect the likelihood of each base being involved in a stable pairing interaction, and not directly a fixed Gibbs free energy (ΔG). Therefore, while red-colored regions generally correspond to highly stable structures with more negative ΔG values, the color mapping is fundamentally based on base-pairing probability rather than absolute thermodynamic thresholds. This allows a probabilistic and visually intuitive interpretation of the structural rigidity and accessibility across the sequence.

Therefore, since this visualization is relative to the ΔG range of the sequence itself, a complementary analysis was performed using the OligoAnalyzer Tool, from IDT (https://www.idtdna.com/calc/analyzer, accessed on 1 April 2025), for double checking and objective quantification of the structures. In OligoAnalyzer^TM^, the predicted hairpin structure with the lowest ΔG was automatically selected as the most stable among the listed possible conformations. The total ΔG of the structure, the melting temperature (Tm), the enthalpy (ΔH) and the entropy (ΔS) were recorded. Furthermore, the contributions of each structural element (such as stacks, loops and helices) were individually analyzed, identifying closing base pairs and specific regions of hairpin formation. The structures were visualized both in list format (with thermoenergetic details) and in two-dimensional graphical representation. The structure considered most relevant for analysis was the one with the greatest potential to interfere in the accessibility of the capture and detection probes, i.e., hairpins that overlap critical hybridization regions.

In OligoAnalyzer™, homodimer prediction for identical ssDNA molecules was evaluated using the following parameters: qPCR mode (default), target type (DNA), oligo concentration (0.2 µM), Na^+^ concentration (50 mM), Mg^2+^ concentration (3 mM), and dNTP concentration (0.8 mM). These parameters are used by the tool to refine thermodynamic estimates, such as melting temperature (Tm), and they are based on specific conditions typical of PCR/qPCR systems. While such settings may influence the calculated ΔG values, we considered that their practical impact on hybridization efficiency under room-temperature conditions, as applied in AF-NALFA, is limited.

### 2.2. Target Genes and Asymmetric PCR

The selection of genetic targets for *L. monocytogenes*, *M. leprae* and *L. amazonensis* was based on genes involved in the expression of virulence factors, which are relevant molecular biomarkers for detecting these pathogens (Table 1). For each pathogen, a primer pair was chosen or designed using the Primer3Plus tool (https://www.primer3plus.com/index.html, accessed on 2 March 2025). These primer pairs, at a 20:1 concentration ratio, were used in an ultra-fast asymmetric PCR (aPCR) with KOD One^®^ Master Mix Blue (KMM-201NV, Sigma-Aldrich, St. Louis, MO, USA) [14] in a final reaction volume of 20 µL, containing 1.0 µL of genomic DNA (gDNA; 20–40 ng/µL) from *M. leprae*, *L. monocytogenes* ATCC 13932 (WDCM 00021 Vitroids™, Millipore, Danvers, MA, USA) and *L. amazonensis* FLA/BR/67/PH8.

The gDNA from *M. leprae* was extracted from mouse footpads using the DNeasy Blood and Tissue Kit (Qiagen, Hilden, Germany), as previously described [25]. The gDNA from the other pathogens was extracted from pure cultures using the GenElute Bacterial Genomic DNA Kit (Sigma-Aldrich), according to the manufacturer’s instructions. The aPCR was carried out with 40 cycles of denaturation at 98 °C for 10 s, annealing (at the target-specific temperature, as shown in Table 1) for 5 s, and extension for 1 s. Amplification products were visualized on 2% (*w*/*v*) agarose gel stained with SYBR Safe^®^ (Thermo Fisher Scientific, Waltham, MA, USA).

### 2.3. Oligonucleotide Probes and Assembly of AF-NALFA Prototypes

AF-NALFA relies on three specific oligonucleotide probes: the detection probe, conjugated to colloidal gold nanoparticles (AuNPs); the capture probe, immobilized at the test line (T), and the control probe, immobilized at the control line (C). To target the genes of interest, detection probes were synthesized with a 3′ thiol modification and a polyT or polyA spacer. Both capture and control probes were biotinylated: the capture probe carried a 5′ biotin with a polyT or polyA spacer, whereas the control probe carried a 3′ biotin with a polyT or polyA spacer (Table 2). All oligonucleotides were synthesized by Sigma-Aldrich. Upon application of the amplicon, it is first hybridized by the detection probe at 5′ or 3′ end and subsequently by the capture probe at the opposite end, producing a visible color signal on the T-line. The assay is validated by the hybridization of the detection probe and the control probe at the C-line

AF-NALFA assembly was performed as described by Lopes-Luz (2025) [14] with modifications. Oligonucleotide probes were resuspended in ultrapure water (Elix^®^, Millipore, Danvers, MA, USA) at a concentration of 100 µM, and 5.0 × 10^−6^ mol/mL CP and ConP were mixed with streptavidin (Sigma-Aldrich) and then coated onto nitrocellulose (NC) (Hi-Flow Plus 180 Membrane Cards, Merck, Danvers, MA, USA) using the XYZ 3050 Dispenser System from BioDot Inc (Irvine, CA, USA). Thiol-modified detection probe was used for conjugation with 40 nm-AuNP, and the AuNP-labeled probe was immediately impregnated onto glass fibers using the XYZ Airjet 3050 Dispenser System. Finally, the glass fibers were inserted into coated cards and cut into 5 mm wide strips (CM4000 BioDot Paper Cutter, Irvine, CA, USA). The strips were inserted into plastic cassettes and stored at 4 °C, protected from light and moisture.

In general, AF-NALFA operates through the sequential hybridization of the AuNP-labeled detection probe to one end of the ssDNA target and the capture probe immobilized at the T-line to the opposite end, generating a visible signal, while the C-line confirms assay validity (Figure 1).

### 2.4. Synthetic ssDNA Constructs

To directly evaluate the influence of *site-specific*-ΔG on the hybridization of capture and detection probes with amplicons in the AF-NALFA, a first panel of synthetic ssDNA was designed and synthesized with intentional modifications in their nucleotide sequences (Appendix A). These alterations were intended to create variations in thermodynamic stability (ΔG) of secondary structures, such as hairpins, homodimers (self-dimers) and other motifs, to better understand their impact on probe hybridization efficiency. The synthetic ssDNA panel included truncated versions of the original amplicons of *L. monocytogenes*, *M. leprae* and *L. amazonensis*, as well as sequences with strategically positioned mismatches and additional complementary regions. These designs were chosen to simulate potential hybridization scenarios and assess the formation of unwanted structures, mainly hairpins and homodimers, which could interfere with assay performance. Appendix A presents the sequences of the native ssDNA amplicon, as well as the synthetic and truncated ssDNA (constructs), along with their respective calculated ΔG values at the hybridization sites of the capture and detection probes.

For *L. monocytogenes*, four synthetic ssDNA variants of the native *hlyA* amplicon were synthesized. *hlyA*-1 maintained the original sequences of the detection and capture probe hybridization sites and served as an unmodified reference. *hlyA*-2 incorporated a random sequence (TATAGGCAATGGG) between the detection and capture hybridization sites, introducing structural complexity. *hlyA*-3 extended this modification by adding a GTTG segment to the inserted sequence, while *hlyA*-4 further included an additional TGTT, leaving only one nucleotide unpaired at the 3′ end within the capture probe binding region.

For *M. leprae*, three synthetic ssDNA variants of the native *RLEP* amplicon (148 bp) were synthesized. *RLEP*-1 maintained the original sequences of the detection and capture probe hybridization sites and served as an unmodified reference. *RLEP*-2 contained a hypothetical single-nucleotide polymorphism (SNP) (A→G) within the capture probe region, while *RLEP*-3 included three hypothetical SNPs: one (C→T) in the detection probe and two (A→G and A→T) in the capture probe, designed to progressively reduce probe-binding complementarity.

For *L. amazonensis*, four synthetic ssDNA were based on the native *ITS1* amplicon. *ITS1*-1 served as the unaltered reference, with hybridization regions of the detection and capture probes unchanged. *ITS1*-2 retained the same probe sites but included a native internal insertion (GATGGA) between them. *ITS1*-3 introduced four substitutions (CGTG) in the detection region and two added adenines (AA) after the capture site, while *ITS1*-4 featured a single substitution (C→T) in the detection region with an unchanged capture site.

Subsequently, a second panel of synthetic constructs of the *L. amazonensis ITS1* gene (*ITS1*-5 to *ITS1*-14) was designed to isolate the effect of *site-specific*-ΔG on probe hybridization independently of sequence mismatches, as all synthetic *ITS1* ssDNA maintained perfect complementarity to both probe-binding sites. Then, synthetic random sequences were inserted at either the 3′ end (*ITS1*-5 to *ITS1*-9; affecting CP) or the 5′ end (*ITS1*-10 to *ITS1*-14; affecting DP) to progressively increase local secondary structure and reduce *site-specific*-ΔG at each binding site (Appendix A). This allowed controlled testing of how many consecutively blocked nucleotides are sufficient to impair T-line signal generation, and whether blocking the hybridization sites of the detection and capture probes results in a differential impact on the performance of AF-NALFA

Regarding the evaluation of the influence of mismatches, a second panel of truncated oligos of *M. leprae RLEP* gene was also designed. *RLEP*-4 was used as a reference because it is fully complementary to detection and capture probes. *RLEP*-5 contained a SNP within the detection probe-binding region, while *RLEP*-5.1 carried the corresponding SNP mirrored in the capture probe-binding region. Likewise, *RLEP*-6 and *RLEP*-6.1 contained three SNPs each, positioned in the binding regions of the detection and capture probe, respectively; *RLEP*-7 and *RLEP*-7.1 contained four SNPs; *RLEP*-8 and *RLEP*-8.1 contained five SNPs, and *RLEP*-9 and *RLEP*-9.1 contained six SNPs in their respective binding regions (Figure 2).

### 2.5. AF-NALFA Execution, Results Registration and Statistical Analysis

The same aPCR amplicons evaluated by 2% agarose gel electrophoresis were used in AF-NALFA for visualization of ssDNA with the naked eye. Then, 3 µL of amplicons were applied with 4*x* SSC running buffer containing 4M urea for 20 min. Using a 48-megapixel camera of the iPhone 15 under the same ambient light conditions, pictures of the T and C lines were taken for quantitative analysis. To analyze the intensities of the T and C lines, we used ImageJ software to evaluate the green channel (split channel) following the protocol of Parolo et al. (2020) [26]. The peak value of the test line (arbitrary unit, a.u.) was obtained by subtracting the background signal from the peak value and then plotting the values on the graph. Statistical analysis was performed using GraphPad Prism version 8.0.1 (https://www.graphpad.com/, accessed on 15 April 2025). The Shapiro–Wilk test was applied to assess the normality of data distribution. Depending on the result, either parametric tests (Student’s t-test, One-way ANOVA) or non-parametric tests (Mann–Whitney U, Kruskal–Wallis) were selected to compare groups. Correlation between hybridization intensity (a.u. mean) and predicted structural parameters (ΔG values) was assessed using Spearman’s rank correlation coefficient. For graphical exploration and evaluation of trends, linear regression analysis was also performed, including estimation of the best-fit line, 95% confidence intervals, R^2^ values, and residual distribution (Runs test) to verify deviation from linearity. All tests were considered statistically significant when *p* < 0.05.

## 3. Results and Discussions

### 3.1. Native ssDNA

The asymmetric PCR (aPCR) amplification of genomic DNA from *M. leprae*, *L. monocytogenes* and *L. amazonensis* yielded products of the expected sizes, as confirmed by 2% (*w*/*v*) agarose gel electrophoresis, with no amplification detected in the no-template control (NTC) (Figure 3a). Then, these aPCR products were applied onto the AF-NALFA prototypes to assess their detection by the naked eye. Visual detection of the tests revealed strong T-line signals for *L. monocytogenes* as expected [14], weak visual T-line signal for *L. amazonensis*, and absence of visual T-line signal for *M. leprae* (Figure 3b, top image). Quantitative analysis showed that the mean intensity of the T-line was highest for *L. monocytogenes* (120.3 ± 18.62 a.u.), followed by a markedly lower intensity for *L. amazonensis* (6.93 ± 1.89 a.u.) and *M. leprae* (1.27 ± 0.80 a.u.), comparable to the blank control (0.65 ± 0.61 a.u.). Statistical comparison confirmed a significant difference between *L. monocytogenes* and the blank (**** *p* < 0.0001), while *L. amazonensis* and *M. leprae* did not differ significantly (Figure 3b, bottom image).

These results indicate that successful aPCR amplification does not necessarily translate into efficient detection of amplicons in AF-NALFA. Although the bands observed in the gel confirm the specific amplification of the three targets, the presence of asymmetric products, with a characteristic ssDNA distribution, should not be interpreted as a definitive indication of adequate device performance, since the behavior of amplicons during hybridization may differ from that observed in electrophoresis. In classical molecular hybridization assays, such as fluorescent in situ hybridization (FISH), it is widely recognized that the formation of secondary structures can render certain target sites inaccessible, resulting in detection failures even in the presence of amplified products [23,27,28]. Thus, the weak or absent T-line signals obtained for *M. leprae* and *L. amazonensis* in AF-NALFA suggest that intrinsic characteristics of these amplicons, such as the local structuring of the probe-binding regions, may be limiting their association with the detection or capture probes. These initial findings reinforce the need to specifically investigate the local thermodynamic determinants that impact the hybridization stage in AF-NALFA systems, which have been poorly explored in the literature.

Next, in-silico analysis of the three pathogen genes was performed to evaluate possible formation of secondary structures that could interfere with hybridization in the AF-NALFA. Hairpin prediction showed that all three native pathogen ssDNA amplicons displayed similar overall structural stability, with optimal overall ΔG values around −35 kcal/mol: *L. monocytogenes hlyA* gene (−33.88 kcal/mol) (Appendix A), *M. leprae RLEP* gene (−35.34 kcal/mol) and *L. amazonensis ITS1* gene (−35.27 kcal/mol) (Appendix A). Although the three amplicons evaluated presented very similar overall ΔG values, their performance in AF-NALFA was significantly different, suggesting that global thermodynamic stability does not explain the observed results. Traditionally, in hybridization assays, probe-target affinity has been interpreted using overall ΔG, which is as the energy balance of the oligo’s association with the target, considering their internal structures [19,21,23,29]. However, despite being widely used to predict hybridization efficiency, these global models failed to capture critical local differences that could interfere with the detection step in AF-NALFA.

Thus, the analysis was directed to evaluate the *site-specific*-ΔG in the regions of interest, revealing that the *M. leprae RLEP* amplicon presented locally more negative values in the capture and detection regions, compatible with the formation of hairpins and homodimers that could block hybridization with the test probes. In hairpin analysis, only *RLEP* exhibited warmer color signals in critical regions of probe hybridization, specifically at the probe detection site (dark orange 0.8–0.9). Detailed folding analysis highlighted a greater extent of structural hindrance at the *RLEP* detection binding site, with 12 nucleotides likely involved in hairpin formation.

Regarding homodimer formation, the *M. leprae RLEP* amplicon exhibited the lowest *site-specific*-ΔG values in both probe regions, indicating a higher thermodynamic propensity for intermolecular pairing. This finding aligns with Li et al. (2005) [28], who demonstrated that ΔG is a reliable predictor of hybrid stability and the tendency to form internal structures. Similarly, Weckx et al. (2007) [21] highlighted the strong correlation between ΔG and binding efficiency in microarrays, supporting the interpretation that lower ΔG values may compromise probe accessibility and hybridization performance. In the detection hybridization region, a *site-specific*-ΔG of −19.64 kcal/mol was observed, involving 7 paired nucleotides (CGGCGGC), while the capture region showed a ΔG of −11.77 kcal/mol for 5 paired nucleotides (Appendix A). The *L. amazonensis ITS1* sequence also showed moderate homodimer potential at the detection probe site (*site-specific*-ΔG = −13.41 kcal/mol, 6 nucleotides), but negligible interaction in the capture probe region (*site-specific*-ΔG > −10 kcal/mol) (Appendix A). In contrast, the *L. monocytogenes hlyA* (Appendix A) amplicons showed minimal likelihood of homodimer formation in both hybridization regions (*site-specific*-ΔG close to or greater than −10 kcal/mol), indicating favorable structural accessibility for probe binding. Notably, these early empirical observations indicated that values around −10 kcal/mol could serve as a practical threshold for discriminating structurally accessible from inaccessible sites in AF-NALFA, a rationale further examined in subsequent analyses.

The detection efficiency of amplicons by AF-NALFA was correlated with the *site-specific*-ΔG in the hybridization regions, so that more negative values suggest lower structural accessibility and a greater propensity for the formation of hairpins or homodimers that hinder probe pairing. In biosensors with similar architecture, Kim et al. (2024) [30] proposed −7.5 kcal/mol as the minimum threshold to ensure efficient hybridization, while Matveeva et al. (2003) [19] reported that self-structures become increasingly stable when ΔG values are more negative than approximately -8 kcal/mol, which can indirectly compromise probe-target hybridization. Based on our empirical results with *L. monocytogenes* we observed that successful detection was consistently associated with *site-specific*-ΔG values close to or greater than −10 kcal/mol. This empirical observation, together with the supporting evidence from previous studies, led us to adopt -10 kcal/mol as a conservative threshold in our analysis. Then, this value was adopted as a practical threshold to indicate sites with satisfactory structural accessibility, since values more negative than this, as observed for *RLEP* and *ITS1*, tend to compromise the probe-target hybridization. Thus, these in silico data provide the first mechanistic support for the experimental results, indicating that local conformational accessibility is crucial for the performance of AF-NALFA to detect amplicons.

To complement the understanding of thermodynamic predictions, Figure 4 illustrates the detailed nucleotide accessibility profiles for the three native pathogen amplicons, highlighting the regions predicted to be structurally accessible (*site-specific*-ΔG > −10 kcal/mol) or involved in secondary structure formation. In the *L. monocytogenes hlyA* amplicon (Figure 4a), both detection and capture regions were fully accessible, with no nucleotides predicted to be involved in secondary structure formation across the entire probe-binding sites. In the *M. leprae RLEP* amplicon (Figure 4b), the detection region exhibited four main sequences of nucleotides involved in secondary structures: one stretch of 7 consecutive blocked nucleotides, followed by additional segments of 2, 2, and 3 nucleotides. Structurally accessible positions were limited to short segments of 2, 1, 1, and 3 consecutive nucleotides. In the capture region, blocking was observed in three sequences of 4 nucleotides each, interspersed with accessible segments of 1, 4, and 3 consecutive nucleotides.

In the *L. amazonensis ITS1* amplicon (Figure 4c), the detection probe region displayed three sequences of 3, 3, and 2 consecutive blocked nucleotides, while accessible regions consisted of sequences of 7 and 5 consecutive nucleotides. For the capture probe region, blocking occurred in 1, 2, and 7 consecutive nucleotides, while accessible sequences included 1, 3, and 6 consecutive nucleotides. This visual analysis confirms that, while *hlyA* exhibits mostly accessible hybridization regions, *RLEP* and *ITS1* contain multiple nucleotides predicted to be structurally constrained, particularly within the detection probe-binding region for *RLEP* and within both detection probe-binding regions for *ITS1*, consistent with the differential hybridization efficiencies observed experimentally (Figure 1).

This visual mapping of native pathogen amplicons reinforces the hypothesis that a *site-specific*-ΔG > −10 kcal/mol serves as a key parameter for effective hybridization in AF-NALFA. The direct correlation between structural accessibility and experimental T-line signal intensity supports this thermodynamic threshold as a reliable predictor of probe-binding efficiency. Guided by these observations, the first panel of truncated oligonucleotides (Appendix A) was designed to further challenge and validate this proposed threshold.

In summary, the analysis of native amplicons demonstrated that, despite successful amplification, only those with structurally accessible probe-binding regions (*site-specific*-ΔG close to or greater than −10 kcal/mol) produced detectable AF-NALFA signals, establishing local thermodynamic accessibility as a critical determinant of assay performance.

### 3.2. Synthetic and Truncated ssDNA

The synthetic and truncated ssDNA (synthetic DNA) *L. monocytogenes hlyA*-1, designed to reduce secondary structure formation at both detection and capture probe sites, showed the highest T-line signal intensity (149.9 ± 10.5 a.u.), significantly exceeding the native *hlyA* ssDNA (120.3 ± 18.6 a.u.,**** *p* < 0.0001). From *hlyA*-2 onward, modifications were introduced to impair hybridization: *hlyA*-2, with a drastic drop in *site-specific*-ΔG at the capture probe site (–52.5 kcal/mol), presented a moderate reduction in signal (64.0 ± 8.1 a.u., **** *p* < 0.0001 vs. *hlyA*-1). Variants *hlyA*-3 and *hlyA*-4, which further intensified interference at both detection and capture probe regions (*site-specific*-ΔG Capture Probe: –69.6 and –80.0 kcal/mol, respectively), resulted in low T-line signal intensities (21.8 ± 7.5 and 28.7 ± 7.9 a.u.), with no significant difference between them (ns), but significantly lower than all other groups (*** *p* < 0.001). No T-line signal was detected in the blank control (Figure 5a and Appendix A).

The superior performance of *hlyA*-1 over the native ssDNA indicates that removing flanking sequences increases the efficiency of amplicon detection by AF-NALFA by reducing structural barriers to hybridization. Native *L. monocytogenes hlyA* ssDNA (207 bp), as a PCR product, can form extensive secondary structures or competing strands, a phenomenon already observed in microarrays and DNA chips, in which long targets exhibit lower hybridization efficiency than shorter synthetic oligos [31,32]. As they contain only ~40 bp critical for probe hybridization, the truncated oligos serve as highly accessible targets for hybridization, allowing the system to reach its full detection potential. In contrast, the progressive T-line signal decrease observed in *hlyA*-2 to *hlyA*-4, concomitant with increasingly negative *site-specific*-ΔG values, reinforces the limiting role of local secondary structures during hybridization [20,33]. These findings experimentally validate *site-specific*-ΔG as a practical predictive parameter and help explain the poorer performance of native targets with higher structural complexity.

Despite the absence of T-line signal for the native *M. leprae RLEP* ssDNA (mean: 1.27 ± 0.80 a.u.), all three synthetic ssDNAs generated visually detectable T-line intensities in the AF-NALFA (Figure 5b and Appendix A). *RLEP*-1, which maintained intact detection and capture hybridization regions, produced the highest average T-line intensity (136.7 ± 8.8 a.u.), followed closely by *RLEP*-2 (131.1 ± 8.3 a.u.), which included a hypothetical SNP (A→G) in the capture probe region. No significant difference was observed between these two constructs (ns). In contrast, *RLEP*-3, carrying three hypothetical SNPs, including one in the detection probe region and two in the capture probe region, showed a significant reduction in T-line signal (114.7 ± 9.5 a.u.), differing from both *RLEP*-1 (**** *p* < 0.0001) and *RLEP*-2 (*** *p* < 0.001). The blank control remained negative (mean: 0.53 ± 0.85 a.u.)

Unlike the native *M. leprae RLEP* ssDNA, which is fully complementary but thermodynamically inaccessible, all truncated constructs exhibited intense T-line signals, demonstrating that structural accessibility is essential for amplicon detection by AF-NALFA. The *RLEP*-2 result suggests that a single mismatch can be tolerated when the *site-specific*-ΔG is greater than −10 kcal/mol, while the decrease in *RLEP*-3 indicates that complementarity becomes relevant with multiple mismatches, a pattern already reported in oligonucleotide-based biosensors, in which the effect of mismatches depends on their number and position [28,34,35,36]. These findings indicate that mismatch tolerance is a secondary factor, which was explored in a controlled manner in the following panel.

Similarly, for the *L. amazonensis ITS1* gene (Figure 5c and Appendix A), the native amplicon generated a weak T-line, whereas all truncated constructs generated strong T-line signals, confirming that restoring structural accessibility is sufficient to enable AF-NALFA detection. As observed for *M. leprae RLEP* gene, constructs with intact probe sites yielded the highest intensities (*ITS1*-1), while additions or substitutions that increased local steric complexity progressively reduced the T-line signal (*ITS1*-2 and *ITS1*-3), although even a single mismatch (*ITS1*-4) could still be tolerated at high intensity levels. Taken together, these data indicate that both thermodynamic accessibility and probe-target complementarity act as critical and possibly synergistic determinants for the successful detection of amplicons by AF-NALFA.

In summary, experiments with synthetic and truncated constructs demonstrated that improving structural accessibility markedly enhances AF-NALFA performance, while progressive structural constraints or multiple mismatches reduce signal intensity. Notably, mismatches at the capture probe site had a stronger impact than those at the detection site, establishing that both thermodynamic accessibility and probe-target complementarity act as predictive parameters for hybridization efficiency.

A second panel of synthetic ssDNA oligos was designed to further investigate the impact of probe-target complementarity and local *site-specific*-ΔG on AF-NALFA performance, using *RLEP* and *ITS1* sequences under controlled structural and thermodynamic variations. The gradual introduction of mirrored SNPs in the detection and capture probe hybridization sites revealed different tolerance thresholds for hybridization efficiency (Figure 6). With a single SNP, no statistical (ns) difference was observed between the detection probe site (127.4 ± 17.1 a.u.) and the capture probe site (131.1 ± 8.27 a.u.). At three SNPs, the capture probe site showed a significant reduction in T-line signal (95.0 ± 20.5 a.u.) compared to the detection probe site (137.9 ± 14.5 a.u.) (**** *p* < 0.0001), and the same trend was seen with four SNPs (capture: 106.3 ± 21.8 a.u.; detection: 131.5 ± 8.74 a.u.) (* *p* < 0.0205). From five SNPs onwards, T-line signal intensity dropped sharply in both sites, with a more pronounced decline in the capture probe site (1.60 ± 1.02 a.u.) than in the detection probe site (55.8 ± 5.66 a.u.) (**** *p* < 0.0001). With six SNPs, both were near baseline (capture: 0.41 ± 0.36 a.u.; detection: 32.1 ± 5.33 a.u.).

These results indicate that AF-NALFA begins to lose hybridization efficiency with approximately 3–4 mismatches, which is a lower tolerance than typically observed in upstream amplification reactions, in which systems like Recombinase Polymerase Amplification (RPA) may accommodate up to 7–9 mismatches (~8–11%) before loss of yield [37]. Unlike conventional or asymmetric, which are highly sensitive to mismatches at the 3′ terminus due to enzyme extension requirements [38], AF-NALFA hybridization relies on passive duplex formation along the length of the oligo [39], making the internal distribution and number of mismatches more determinant than their terminal position.

The greater reduction in AF-NALFA T-line signal caused by mismatches at the capture probe site, relative to the detection probe site, can be attributed to several physicochemical factors. First, the capture probe is immobilized onto a nitrocellulose membrane, requiring the duplex to be stable under the capillary flow of the strip, whereas the detection probe-target interaction starts when the sample migrates through the conjugated pad by capillarity, before reaching the nitrocellulose membrane, and thus benefits from greater conformational flexibility and longer contact time in solution. This is consistent with findings by Pesciotta et al. (2011) [40], which demonstrated that surface-immobilization significantly perturbs hybridization kinetics, increasing *K*d values by ~50% compared to free-solution interactions. Also, Sohreiner et al. (2011) [41] showed that oligonucleotides conjugated to AuNPs adopt upright conformations and favorable spacing that benefits efficient hybridization, while immobilized capture probe may be more sterically restricted and sensitive to base-pair disruptions. Studies by Peytavi et al. (2005) [42] and Macedo et al. (2017) [43] highlight that probe position relative to the target, spacing, and surface density strongly influence hybridization efficiency due to secondary structure formation and steric crowding effects. Finally, Doria et al. (2010) [34] demonstrated that mismatch localization and thiol-oligonucleotide density on AuNPs affect mismatch discrimination, with 3′-end mismatches on AuNP probes being better tolerated.

From a biological and diagnostic perspective, these findings are especially relevant for targets prone to genetic variability or hypermutation. In such scenarios, mapping mismatch tolerance is essential to anticipate potential losses in sensitivity and to design detection strategies that remain effective without prior knowledge of variant identity [36,44]. This is particularly important for AF-NALFA applications in point-of-care settings and for pathogen surveillance, where target sequences may contain previously uncharacterized SNPs, but robust detection is still required.

Finally, a new panel of synthetic constructs of the *ITS1* gene of *L. amazonensis* was designed to assess the influence of progressively lower *site-specific*-ΔG values on the generation of the AF-NALFA T-line signal under conditions of full probe-target complementarity (i.e., in the absence of mismatches) (Figure 7). For capture probe-targeted constructs (*ITS1*-5 to *ITS1*-9), *ITS1*-5 (*site-specific*-ΔG = −9.15 kcal mol^−1^; 7 nucleotides) generated a T-line signal of 150.0 ± 17.08 a.u., *ITS1*-6 (*site-specific*-ΔG = −10.63 kcal mol^−1^; 8 nucleotides) yielded 158.7 ± 4.12 a.u., and *ITS1*-7 (*site-specific*-ΔG = −17.22 kcal mol^−1^; 12 nucleotides) produced 142.2 ± 7.12 a.u. T-line signal intensity then declined progressively in *ITS1*-8 (*site-specific*-ΔG = −22.24 kcal mol^−1^; 14 nucleotides), which showed 127.6 ± 8.82 a.u., and *ITS1*-9 (*site-specific*-ΔG = −32.37 kcal mol^−1^; 19 nucleotides), which registered 108.9 ± 8.70 a.u.; both being significantly lower than *ITS1*-5 (*p* = 0.0011 and *p* < 0.0001, respectively). Curiously, the *ITS1*-9 construct, which retained only a single potentially accessible nucleotide at the end of the capture probe site (*site-specific*-ΔG > −10 kcal/mol), still yielded a T-line signal intensity above 100 a.u., reinforcing the high tolerance of the AF-NALFA detection system to extensive secondary structure formation when probe–target complementarity is preserved (Figure 7).

Among detection probe-targeted constructs (*ITS1*-10 to *ITS1*-14), *ITS1*-10 (*site-specific*-ΔG = −13.51 kcal mol^−1^; 6 nucleotides) presented the highest T-line signal (174.5 ± 6.61 a.u.), followed by *ITS1*-11 (*site-specific*-ΔG = −12.63 kcal mol^−1^; 8 nucleotides) and *ITS1*-12 (*site-specific*-ΔG = −15.55 kcal mol^−1^; 9 nucleotides), which yielded 150.9 ± 20.9 and 132.8 ± 10.76 a.u., respectively. *ITS1*-13 (*site-specific*-ΔG = −18.84 kcal mol^−1^; 11 nucleotides) and *ITS1*-14 (*site-specific*-ΔG = −32.25 kcal mol^−1^; 17 nucleotides) resulted in high intensities of 152.4 ± 2.97 a.u. and 153.2 ± 5.60 a.u., respectively. Within this set, *ITS1*-12 was significantly lower than *ITS1*-10 (*p* < 0.0001) and *ITS1*-13 (*p* = 0.0078). Despite these differences, all constructs generated visually strong T-line signals above 100 a.u., regardless of the degree of secondary structure predicted for each site (Figure 7).

Although progressively negative *site-specific*-ΔG values were associated with reduced AF-NALFA T-line signal intensities, the fact that synthetic oligos with *site-specific*-ΔG < −20 kcal/mol still produced visible T-lines above 100 a.u. suggests a partial structural tolerance under highly controlled conditions and perfect complementarity. Nonetheless, in native amplicon contexts, in which additional structural constraints are present, the *site-specific*-ΔG remains one of the deterministic parameters for detection success, highlighting that thermodynamic accessibility should be prioritized during probe and target design. Moreover, the absence of total T-line signal loss at either the detection or capture probe sites, despite the introduction of strong structural hindrance, reinforces that the dual-probe “sandwich” architecture offers an inherent kinetic robustness. In this configuration, partial impairment at one probe-binding site may be compensated by stable hybridization at the other, allowing sufficient retention of the complex for visual detection under capillary flow. A similar notion of kinetic robustness has also been reported in modified lateral flow architectures, in which pre-hybridization in solution significantly enhances T-line signal retention and detection performance [45], supporting the idea that stable interactions at one probe site can compensate for partial impairments at another under flow conditions.

In the context of *M. leprae* DNA detection, the multicopy *RLEP* gene is a valuable amplification target for diagnostics due to its high genomic copy number, which theoretically enhances sensitivity [46,47]. However, our AF-NALFA data demonstrate that its repetitive nature also promotes the formation of stable secondary structures that elevate the *site-specific*-ΔG and hinder probe accessibility. Remarkably, even with full probe–target complementarity, the native *RLEP* amplicon failed to hybridize, indicating that thermodynamic inaccessibility acts as a decisive barrier in heterogeneous samples enriched with diverse genomic material. Hence, while repetitive targets improve upstream amplification, they may paradoxically compromise downstream lateral flow detection, emphasizing the importance of ΔG-driven design criteria when applying AF-NALFA for *M. leprae* diagnosis.

For *L. amazonensis*, we used the *ITS1* region, a common diagnostic target due to its interspecies variability, particularly for differentiation from closely related taxa like *L. braziliensis* [48]. Nevertheless, this locus is inherently AT-rich and prone to extensive DNA hairpin and dimer formation [49], which increases local *site-specific*-ΔG and restricts probe accessibility within the AF-NALFA detection format. While synthetic *ITS1* constructs maintained T-line signal despite highly negative *site-specific*-ΔG values under perfect complementarity, the native amplicon failed to generate a detectable T-line signal, highlighting the functional impact of structural complexity in real genomic contexts. This finding reinforces the lesson that thermodynamic determinants must be balanced with biological variability when translating AF-NALFA development to field diagnostics especially for structurally complex parasites, such as *Leishmania* spp.

In summary, while synthetic oligos revealed that AF-NALFA can tolerate substantial secondary structures or even highly negative *site-specific*-ΔG values under controlled conditions with perfect complementarity, the performance with native amplicons from *M. leprae* and *L. amazonensis* highlighted that genomic context and thermodynamic inaccessibility are decisive barriers for detection. Therefore, rational AF-NALFA design requires case-by-case thermodynamic evaluation of probe-target pairs, as no universal ΔG threshold applies uniformly across different pathogens.

Several signal amplification methods in nucleic acid lateral flows have been described, including the use of different nanoparticle sizes and architectures [50,51], biotin–streptavidin coupling systems [52], and advanced plasmonic approaches such as surface-enhanced Raman scattering (SERS) [5]. These strategies can markedly increase visual detection sensitivity, but their analytical performances still depend on efficient probe-target hybridization. Our findings therefore suggest that thermodynamic determinants and signal amplification methods should be regarded as synergistic rather than alternative pathways for assay optimization, with proper probe design providing the foundation upon which amplification strategies can further enhance signal output. However, the analytical performance of these amplification methods cannot be directly compared with the present approach, as they represent distinct directions in assay development.

This study has some inherent limitations. First, each pathogen genome and amplification product has unique thermodynamic determinants, making it impossible to fully replicate the structural and contextual complexity of native amplicons using synthetic oligonucleotides. Even in controlled scenarios with sequence-directed design, mismatch introduction, and *site-specific*-ΔG modulation, synthetic constructs only approximate the physicochemical environment of real amplification products, influenced by salts, buffer components, and genomic context, but never faithfully reproduce it. Furthermore, attempts to create mirrored thermodynamic conditions at the hybridization sites of detection and capture probes are limited by the intrinsic nature of nucleic acids: a single-nucleotide change, insertion, or position shift can unpredictably alter the local thermodynamic profile, preventing the creation of perfectly balanced or symmetrical conditions for hybridization. Consequently, each molecular target for diagnosis must be evaluated individually, as no universal thermodynamic threshold is likely to apply to different genomes. For the practical development of AF-NALFA prototypes, it is essential to perform in silico thermodynamic evaluations of each candidate probe-target pair on a case-by-case basis, considering the specific genomic structural architecture and amplification context of each pathogen.

## 4. Conclusions

Our results demonstrate that the performance of antibody-free nucleic acid lateral flow assays (AF-NALFA) is dictated by a delicate interplay between the thermodynamic accessibility of specific sites in the hybridization regions of the detection and capture probes and the level of sequence complementarity. Even in the absence of mismatches, we show that local structural constraints reflected in highly negative ΔG values for each site can deterministically impair hybridization in amplification products generated by asymmetric PCR, so-called native ssDNA, particularly in repetitive or AT-rich genomic regions, such as those of *M. leprae* and *L. amazonensis*. Conversely, synthetic oligonucleotide constructs revealed that AF-NALFA can tolerate considerable structural hindrance when thermodynamically favorable sites remain accessible, reinforcing that ΔG-driven parameters are effective predictors of hybridization efficiency and of amplicon detection by AF-NALFA. Taken together, these findings highlight that no universal threshold applies uniformly to all pathogens, as different genomic contexts have a substantial influence on detection success. Therefore, the development of specific and sensitive AF-NALFA for diagnostic applications must include a rational, target-specific in silico thermodynamic assessment of probe-target interactions. Future studies should expand this approach to the detection of amplicons from other microorganisms and integrate structural modeling with probe design pipelines to accelerate the development and implementation of rapid, sensitive, and reliable molecular diagnostics at the point of care.

## Figures and Tables

**Figure 1 biomolecules-15-01404-f001:**
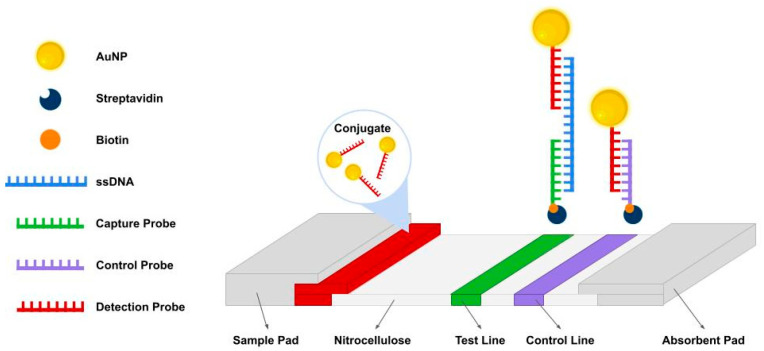
Schematic representation of the antibody-free nucleic acid lateral flow assay (AF-NALFA). The system consists of a sample pad, conjugated pad, a nitrocellulose membrane (NC) with a test line (T) and a control line (C), and an absorbent pad. Detection probes (red) conjugated to AuNPs hybridize with single-stranded amplicons (blue), which are subsequently captured by immobilized probes at the T-line (green). The C-line (purple) ensures proper flow and assay validity. Biotin (orange) and streptavidin (dark blue) mediate the binding of probes to NC.

**Figure 2 biomolecules-15-01404-f002:**
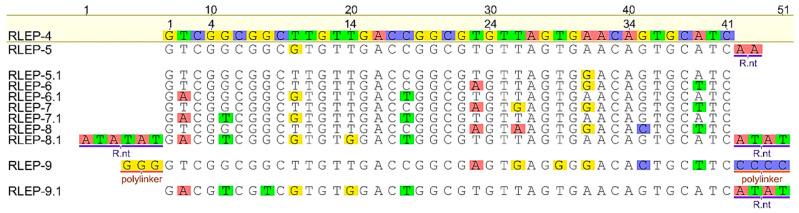
Multiple alignment (Map to Reference) of synthetic truncated ssDNA sequences (*M. leprae RLEP*-4 to *RLEP*-9.1) of the hybridization sites of the detection and capture probes by Geneious^®^. The native reference sequence (*RLEP*-4) is shown at the top. Colored boxes indicate the positions of the hypothetical introduced SNPs. For each construct containing SNPs in the detection region, a corresponding mirrored construct was designed with the same number and position of SNPs in the capture probe region. Guanine (yellow) or cytosine (blue) polylinker and random nucleotides (R.nt) were added to decrease the influence of thermodynamic parameters.

**Figure 3 biomolecules-15-01404-f003:**
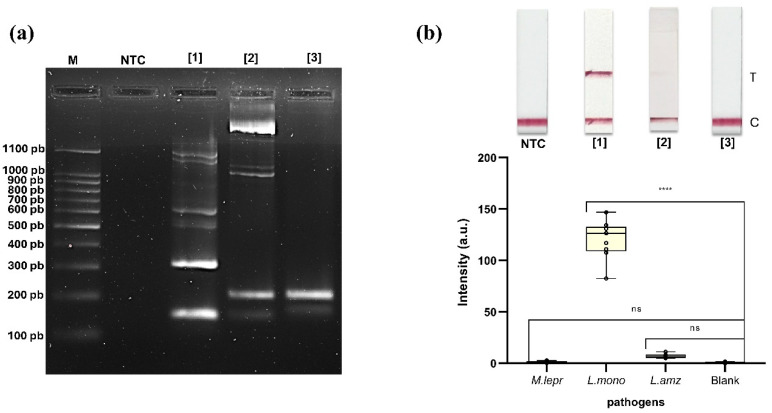
Amplification and detection of native ssDNA of pathogens. (**a**) Asymmetric PCR (aPCR) products obtained from genomic DNA of each pathogen. Specific bands were observed for *M. leprae* (148 bp) [1], *L. monocytogenes* (207 bp) [2] and *L. amazonensis* (205 bp) [3] with no amplification in the no-template control (NTC). Additional bands of other molecular sizes are consistent with ssDNA from asymmetric amplification. M: molecular marker. (**b**) AF-NALFA visual detection and T-line signal quantification. Top: representative strips showing the test line (T) and control line (C) for each pathogen. Bottom: All box plots represent nine replicates, with central lines indicating the median, and whiskers corresponding to minimum and maximum values. a.u.: arbitrary unit; ns: not significant; **** = *p* < 0.0001.

**Figure 4 biomolecules-15-01404-f004:**
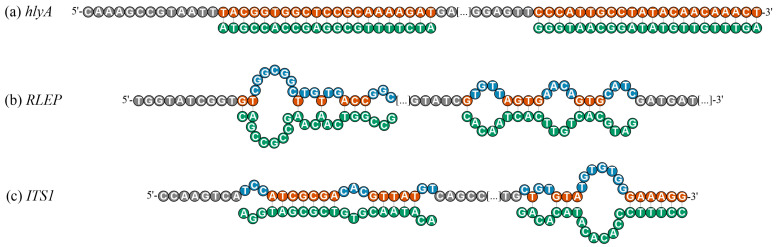
Structural mapping representation of structurally accessible and hindered nucleotides within the hybridization regions of native ssDNA amplicons for the targets (**a**) *hlyA* of *L. monocytogenes*, (**b**) *RLEP* of *M. leprae*, and (**c**) *ITS1* of *L. amazonensis*. The hybridization sites of the detection probe (DP, 5′ region) and capture probe (CP, 3′ region) are annotated in green below each sequence. Nucleotides highlighted in orange indicate bases predicted to remain structurally accessible for hybridization (not involved in hairpin or homodimer formation; site-specific ΔG > −10 kcal/mol). Nucleotides highlighted in blue represent bases predicted to participate in secondary structures, either hairpins (probability ≥ 0.7) or homodimers (site-specific ΔG < −10 kcal/mol). These visual predictions were generated based on Geneious^®^ and OligoAnalyzer™ in silico analyses under hybridization conditions (25 °C).

**Figure 5 biomolecules-15-01404-f005:**
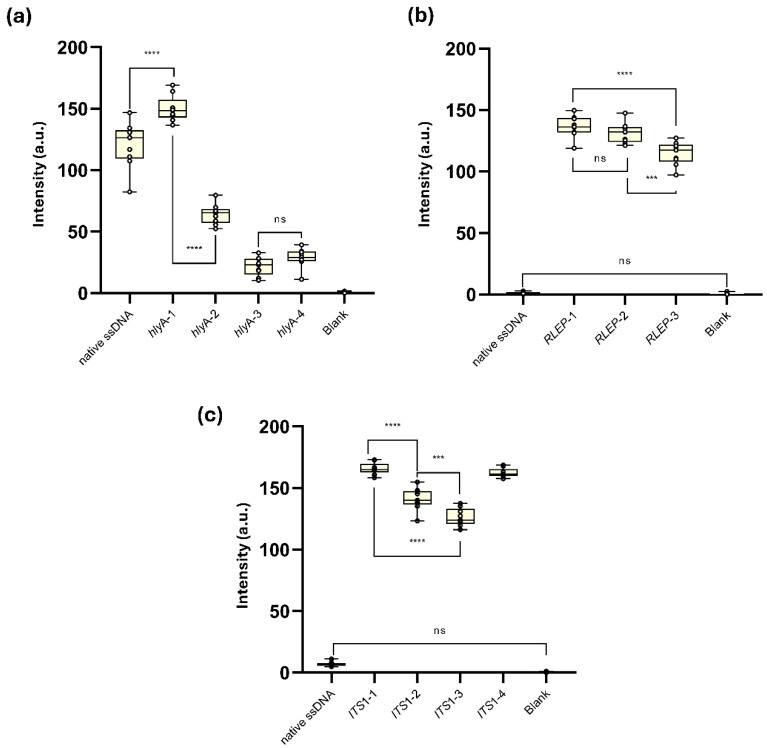
Evaluation of synthetic truncated single-stranded DNA (syn-trunc-ssDNA) constructs designed from the target genes of (**a**) *L. monocytogenes*, (**b**) *M. leprae* and (**c**) *L. amazonensis*, and their performance in AF-NALFA. All box plots represent nine replicates, with central lines indicating the median, and whiskers corresponding to minimum and maximum values. The most relevant values are shown in the plots. Complete numerical data are available in Appendix A. Images of the tests performed are available in Appendix A. a.u.: arbitrary unit; ns: not significant; **** = *p* < 0.0001; *** = *p* < 0.001.

**Figure 6 biomolecules-15-01404-f006:**
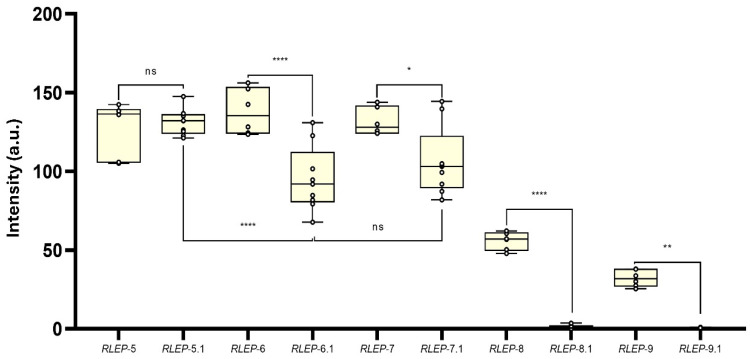
Effect of mirrored SNPs in the detection and capture probe hybridization sites on AF-NALFA T-line signal intensity. Boxplots represent the distribution of T-line signal values (arbitrary units, a.u.) for constructs containing 1, 3, 4, 5, or 6 SNPs in either the detection or capture probe hybridization site. Statistical comparisons were performed between corresponding detection and capture sites for each SNP group, and between selected groups within each site. All data are presented as box plots with new replicates for capture probe and six replicates for detection probe, showing the median, minimum and maximum values. Significance levels: ns: not significant; * *p* < 0.05; ** *p* < 0.01; **** *p* < 0.0001. a.u.: arbitrary unit; ns: not significant.

**Figure 7 biomolecules-15-01404-f007:**
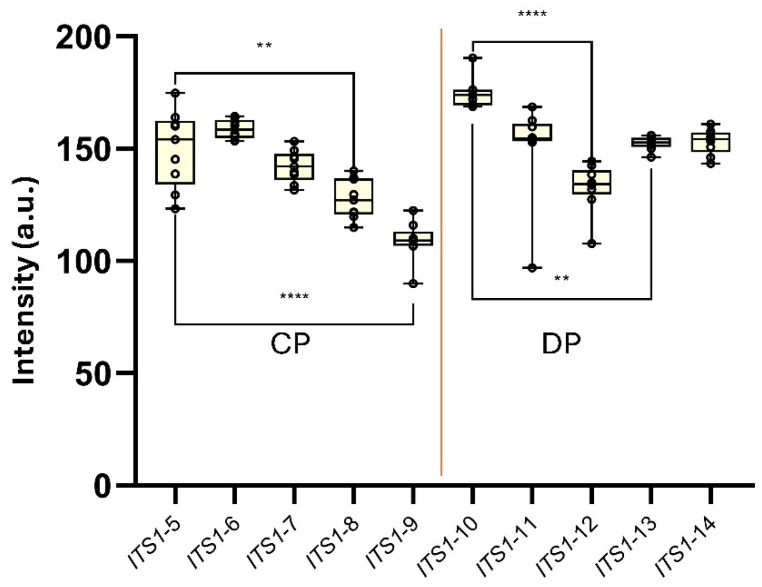
AF-NALFA T-line signal intensities generated by a panel of synthetic *ITS1* ssDNA constructs carrying progressively more negative *site-specific*-ΔG values at either the capture probe site (CP; *ITS1*-5 to *ITS1*-9) or the detection probe site (DP; *ITS1*-10 to *ITS1*-14). Box-and-whisker plots represent the distribution of T-line intensities (a.u.) for each construct. Statistical comparisons were performed using Tukey’s test (**, *p* < 0.01; ****, *p* < 0.0001).

**Table 1 biomolecules-15-01404-t001:** Pathogens, primer sequences (forward and reverse), amplicon sizes, target genes, annealing temperatures, and references. The primers were designed or selected for the detection of *L. monocytogenes*, *M. leprae* and *L. amazonensis*.

Pathogen	Primer Sequences (5′-3′)	Amplicon Size	Gene	Annealing Temperature	References
*L. monocytogenes*	F: CCGTAAGTGGGAAATCTGTCTC	207 bp	*hlyA*	56.5 °C	[6]
	R: AGTTTGTTGTATAGGCAATGGG				
*M. leprae*	F: ATTTCTGCCGCTGGTATCGGT	148 bp	*RLEP*	62.5 °C	[25]
	R: TGCGCTAGAAGGTTGCCGTAT				
*L. amazonensis*	F: CCTTTCCCACACATACACAGC	222 bp	*ITS1*	58.3 °C	This study
	R: ACGAAATAGGAAGCCAAGTCA				

F: forward; R: reverse.

**Table 2 biomolecules-15-01404-t002:** Pathogens, names of probes, oligonucleotide sequences, and respective terminal modifications. The Detection Probes (DP) have 3′-terminal thiol groups and polyT or polyA spacers. Capture Probes (CP) and Control Probes (ConP) are biotinylated at the 5′ or 3′ ends, respectively.

Pathogen	Probe	Sequence
*L. monocytogenes*	DP	5′-ATCTTTTGCGGAGCCACCGTATTTTTTTTTT-[thiol]-3′
	CP	5′-[biotin]-TTTTTTTTTTAGTTTGTTGTATAGGCAATGGG-3′
	ConP	5′-ACGGTGGCTCCGCAAAAGATTTTTTTTTTTT-[biotin]-3′
*M. leprae*	DP	5′-GCCGGTCAACAAGCCGCCGACTTTTTTTTTTTTTTTTTTTT-[thiol]-3′
	CP	5′-[biotin]-TTTTTTTTTTTTTTTTTTTTGATGCACTGTTCACTAACAC-3′
	ConP	5′-GTCGGCGGCTTGTTGACCGGC-TTTTTTTTTTTTTTTTTTTT-[biotin]-3′
*L. amazonensis*	DP	5′-ACATAACGTGTCGCGATGGAAAAAAAAAAA-[thiol]-3′
	CP	5′-[biotin]-AAAAAAAAAAAGCAAGCCTTTCCCACAGAT-3′
	ConP	5′-TCCATCGCGACACGTTATGTAAAAAAAAAA-[biotin]-3′

## Data Availability

The original contributions presented in this study are included in the article/Appendix A. Further inquiries can be directed to the corresponding author.

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
