# Peer review of "Thermodynamic Determinants in Antibody-Free Nucleic Acid Lateral Flow Assays (AF-NALFA): Lessons from Molecular Detection of Listeria monocytogenes, Mycobacterium leprae and Leishmania amazonensis"

_biomolecules, 2025, doi:10.3390/biom15101404_

Round 1

Reviewer 1 Report

Comments and Suggestions for Authors

This paper investigates the site-specific thermodynamics (ΔG), local secondary structures and mismatches in the probe binding regions and their impact on the performance of antibody-free nucleic acid lateral flow assay (AF-NALFA) for native and synthetic amplicons. This study will be useful for the development of effective assays for pathogen DNA detection. Authors are encouraged to consider the comments below before recommending this manuscript for acceptance to Biomolecules.

  1. Materials and methods. Section 2.3. Oligonucleotide probes and assembly of AF-NALFA prototypes. In the caption to Figure 1, the authors present the operating principle of AF-NALFA, which makes it difficult to understand. It is recommended to move this text to the results and discussion.
  2. Line 384. “Together, these findings support the adoption of −10 kcal/mol as a conservative threshold in our analysis”. It is not clear from the studies cited previously in the manuscript why the authors chose -10 kcal/mol as the threshold value for ΔG. Please explain the chosen cut-off.
  3. Figure 5. Authors are encouraged to provide strip images with data reflecting the performance of each target gene in AF-NALFA.
  4. In Figure 4, the amino acid sequence is not discernible. The authors are advised to increase the size and improve the quality of the image.
  5. Authors are encouraged to consider including a brief summary for each subsection in the results and discussion, highlighting the key findings.

Author Response

Reviewer 1 – General Comment

This paper investigates the site-specific thermodynamics (ΔG), local secondary structures and mismatches in the probe binding regions and their impact on the performance of antibody-free nucleic acid lateral flow assay (AF-NALFA) for native and synthetic amplicons. This study will be useful for the development of effective assays for pathogen DNA detection. Authors are encouraged to consider the comments below before recommending this manuscript for acceptance to Biomolecules.

Response: We sincerely thank the Reviewer for the positive and encouraging comments on our work. We are pleased that the study was considered useful for the development of effective assays for pathogen DNA detection. We carefully addressed all the points raised and revised the manuscript, accordingly, as detailed in the specific responses below.

Comment 1

  1. Materials and methods. Section 2.3. Oligonucleotide probes and assembly of AF-NALFA prototypes. In the caption to Figure 1, the authors present the operating principle of AF-NALFA, which makes it difficult to understand. It is recommended to move this text to the results and discussion.

Response: We thank the reviewer for this suggestion and agree with the comment. To facilitate understanding, we have moved the description of the operating principle of AF-NALFA from the caption to the main text, placing it immediately before Figure 1 in the Methods section. The caption has been adjusted accordingly (lines 237-242).

Comment 2

Line 384. “Together, these findings support the adoption of −10 kcal/mol as a conservative threshold in our analysis”. It is not clear from the studies cited previously in the manuscript why the authors chose −10 kcal/mol as the threshold value for ΔG. Please explain the chosen cut-off.

Response: We thank the reviewer for this observation. We have revised the text to clarify that the threshold value of −10 kcal/mol was derived empirically from the successful Listeria monocytogenes assay and subsequently discussed in the context of the literature (lines 407–410 and 418-427).

Comment 3

  1. Figure 5. Authors are encouraged to provide strip images with data reflecting the performance of each target gene in AF-NALFA.

Response: We thank the reviewer for this suggestion. Strip images reflecting the performance of each target gene in AF-NALFA have been provided as Supplementary Material.

Comment 4

  1. In Figure 4, the amino acid sequence is not discernible. The authors are advised to increase the size and improve the quality of the image.

Response: We thank the reviewer for this observation and agree with the suggestion. The nucleotide sequences in Figure 4 has been enlarged and the image quality has been improved as requested.

Comment 5

Authors are encouraged to consider including a brief summary for each subsection in the results and discussion, highlighting the key findings.

Response: We appreciate this valuable suggestion. A brief summary highlighting the key findings has been added at the end of each subsection in the Results and Discussion. Please, see lines 457-460; 527-532; 674-680

Reviewer 2 Report

Comments and Suggestions for Authors

This work reported on antibody-free nucleic acid lateral flow assays of DNA. The authors suggest that the novelty of this work is “antibody-free” method. However, most of the reported methods for DNA detection were dependent upon hybridization reactions and did not require the use of antibody. I do not think the “antibody-free” concept is the novelty of this work. Comments:

  1. The general strategies for nucleic acid lateral flows and their progress should be carefully discussed in Introduction.
  2. The signal amplification methods in nucleic acid lateral flows should be discussed and their analytical performances should be compared with this method.
  3. How about the selectivity of this method?
  4. How to control the hybridization temperature in the nucleic acid lateral flow assays?
  5. The references are too old, which should be updated to cite those in recent five years.
  6. What are thermodynamic determinants? Their advantages should be briefly commented in Introduction.

Author Response

Reviewer 2 – General Comment

This work reported on antibody-free nucleic acid lateral flow assays of DNA. The authors suggest that the novelty of this work is “antibody-free” method. However, most of the reported methods for DNA detection were dependent upon hybridization reactions and did not require the use of antibody. I do not think the “antibody-free” concept is the novelty of this work.

Response: We thank the reviewer for this observation. We understand that the way the text was written may have given the impression that the “antibody-free” format was presented as the novelty of this work. We have revised the Abstract and Introduction to clarify that the novelty lies in the systematic evaluation of site-specific Gibbs free energy.

Comment 1

  1. The general strategies for nucleic acid lateral flows and their progress should be carefully discussed in Introduction.

Response: We thank the reviewer for this valuable suggestion. A brief discussion on the general strategies for nucleic acid lateral flow assays and their progress has been added in the Introduction. Please, see lines: 58 a 69

Comment 2

  1. The signal amplification methods in nucleic acid lateral flows should be discussed and their analytical performances should be compared with this method.

Response: We thank the reviewer for this suggestion. The signal amplification methods in nucleic acid lateral flows and their analytical performances have now been briefly discussed.  Please, see lines 681-691

  1. How about the selectivity of this method?

Response: We thank the reviewer for this question. However, the selectivity of the method could not be evaluated within the scope and objectives of the present study.

  1. How to control the hybridization temperature in the nucleic acid lateral flow assays?

Response: We thank the reviewer for this question. In AF-NALFA, the hybridization temperature is controlled by the running buffer rather than by the melting temperature (Tm). This information was added to the Introduction (lines 87–91).

Comment 5

  1. The references are too old, which should be updated to cite those in recent five years.

Response: We thank the reviewer for this observation. We have updated the references to include recent works from the last five years, while also retaining pioneering studies that remain essential to support the context of this work .

  1. What are thermodynamic determinants? Their advantages should be briefly commented in Introduction.

Response: We thank the reviewer for this comment. We have defined the thermodynamic determinants (ΔG, Tm, GC content, secondary structures, and mismatches) and briefly commented on their advantages in the Introduction (lines 80–92).

Round 2

Reviewer 1 Report

Comments and Suggestions for Authors

The authors took into account all the reviewers' comments and significantly improved the manuscript.

Reviewer 2 Report

Comments and Suggestions for Authors

Accept